# Enhancing Neoadjuvant Virotherapy’s Effectiveness by Targeting Stroma to Improve Resectability in Pancreatic Cancer

**DOI:** 10.3390/biomedicines12071596

**Published:** 2024-07-18

**Authors:** Khandoker Usran Ferdous, Mulu Z. Tesfay, Aleksandra Cios, Randal S. Shelton, Conner Hartupee, Alicja Urbaniak, Jean Christopher Chamcheu, Michail N. Mavros, Emmanouil Giorgakis, Bahaa Mustafa, Camila C. Simoes, Isabelle R. Miousse, Alexei G. Basnakian, Omeed Moaven, Steven R. Post, Martin J. Cannon, Thomas Kelly, Bolni Marius Nagalo

**Affiliations:** 1Department of Pathology, University of Arkansas for Medical Sciences, Little Rock, AR 72205, USA; kferdous@uams.edu (K.U.F.); mztesfay@uams.edu (M.Z.T.); acios@uams.edu (A.C.); ccsimoes@uams.edu (C.C.S.); spost@uams.edu (S.R.P.); kellythomasj@uams.edu (T.K.); 2Winthrop P. Rockefeller Cancer Institute, University of Arkansas for Medical Sciences, Little Rock, AR 72205, USA; mmavros@uams.edu (M.N.M.); cannonmartin@uams.edu (M.J.C.); 3College of Medicine, University of Arkansas for Medical Sciences, Little Rock, AR 72205, USA; sheltonrandals@uams.edu; 4Division of Surgical Oncology, Department of Surgery, Louisiana State University (LSU) Health, New Orleans, LA 70112, USA; chartu@lsuhsc.edu (C.H.); omoave@lsuhsc.edu (O.M.); 5Department of Biochemistry and Molecular Biology, University of Arkansas for Medical Sciences, Little Rock, AR 72205, USA; aurbaniak@uams.edu (A.U.); iracinemiousse@uams.edu (I.R.M.); 6Department of Biological Sciences and Chemistry, Southern University and A&M College, Baton Rouge, LA 70813, USA; jeanchristopher.c@sus.edu; 7Division of Biotechnology and Molecular Medicine, Department of Pathobiological Sciences, School of Veterinary Medicine, Louisiana State University, Baton Rouge, LA 70803, USA; 8Department of Surgery, University of Arkansas for Medical Sciences, Little Rock, AR 72205, USA; egiorgakis@uams.edu; 9Department of Pharmaceutical Sciences, University of Arkansas for Medical Sciences, Little Rock, AR 72205, USA; bmustafa@uams.edu; 10Department of Pharmacology and Toxicology, University of Arkansas for Medical Sciences, Little Rock, AR 72205, USA; basnakianalexeig@uams.edu; 11Central Arkansas Veterans Healthcare System, John L. McClellan Memorial VA Hospital, Little Rock, AR 72205, USA; 12Department of Interdisciplinary Oncology, Louisiana Cancer Research Center, Louisiana State University (LSU) Health, New Orleans, LA 70112, USA; 13Department of Microbiology and Immunology, University of Arkansas for Medical Sciences, Little Rock, AR 72205, USA

**Keywords:** borderline resectable, locally advanced, pancreatic ductal adenocarcinoma, oncolytic virus, proteolytic enzymes, stroma degradation, surgical resection, resection margin

## Abstract

About one-fourth of patients with pancreatic ductal adenocarcinoma (PDAC) are categorized as borderline resectable (BR) or locally advanced (LA). Chemotherapy and radiation therapy have not yielded the anticipated outcomes in curing patients with BR/LA PDAC. The surgical resection of these tumors presents challenges owing to the unpredictability of the resection margin, involvement of vasculature with the tumor, the likelihood of occult metastasis, a higher ratio of positive lymph nodes, and the relatively larger size of tumor nodules. Oncolytic virotherapy has shown promising activity in preclinical PDAC models. Unfortunately, the desmoplastic stroma within the PDAC tumor microenvironment establishes a barrier, hindering the infiltration of oncolytic viruses and various therapeutic drugs—such as antibodies, adoptive cell therapy agents, and chemotherapeutic agents—in reaching the tumor site. Recently, a growing emphasis has been placed on targeting major acellular components of tumor stroma, such as hyaluronic acid and collagen, to enhance drug penetration. Oncolytic viruses can be engineered to express proteolytic enzymes that cleave hyaluronic acid and collagen into smaller polypeptides, thereby softening the desmoplastic stroma, ultimately leading to increased viral distribution along with increased oncolysis and subsequent tumor size regression. This approach may offer new possibilities to improve the resectability of patients diagnosed with BR and LA PDAC.

## 1. Introduction

Pancreatic cancer ranks third among all cancer-related fatalities in the United States, with a 5-year relative survival rate of only 12.8% [1,2]. Within the spectrum of pancreatic neoplasms, pancreatic ductal adenocarcinoma (PDAC) constitutes over 90% of all cases, with Kirsten rat sarcoma viral oncogene homolog (KRAS) mutations detected in nearly 9 out of 10 PDAC instances [3,4]. However, the involvement of PI3K/AKT/mTOR, NF-κB, JAK/STAT, and Wingless/nt1 (WNT) pathways has also been associated with the development and progression of PDAC [5]. PDAC originates from ductal epithelial cells in the pancreas and is characterized by features that hinder successful treatment, including elevated interstitial fluid pressure, metabolic reprogramming, the presence of a dense stroma in the desmoplastic tumor microenvironment (TME), immunosuppression within the TME, and inherent heterogeneity [6]. 

The immunosuppressive nature of the TME is characterized by a desmoplastic reaction of the stroma which is influenced by heterogenous cell types like pancreatic stellate cells (PSCs), tumor-associated macrophages (TAMs), cancer-associated fibroblasts (CAFs), regulatory T cells (Tregs), myeloid-derived suppressor cells (MDSCs), and extracellular matrix components like hyaluronic acid, collagen, and decorin [7,8,9]. The TME of PDAC is dominated by the accumulation of cells with immunosuppressive functions in addition to the absence of effector tumor-infiltrating T lymphocytes (TILs). Furthermore, PDAC tumors rarely express neoantigens [10]. These attributes of PDAC TME limit the efficient delivery and efficacy of systemic therapies, including immunotherapies, in stroma-dense PDAC [11]. Preclinical and clinical data indicate that accumulation of hyaluronic acid and collagen, major acellular components of the stroma in PDAC, is associated with aggressive metastatic disease, drug resistance, and poor prognosis [12]. The stiffness of the tumor microenvironment (TME) in PDAC is primarily imparted by collagen, while hyaluronic acid, a large glycosaminoglycan, contributes to increased interstitial fluid pressure by retaining water [13].

Only 20% of patients with PDAC present with surgically resectable local disease, and around 50% of all patients with PDAC are diagnosed with metastasis [14,15]. The remaining 25–30% of patients are categorized as having either borderline resectable (BR) or locally advanced (LA) PDAC. BR PDAC is defined by the presence of microscopic cancer cells in the primary tumor region with limited involvement of surrounding vasculature. Although there was no universally harmonized definition for BR PDAC until 2016, some of its characteristics were no distant metastatic lesion and narrowing of the venous lumen with involvement of the superior mesenteric and portal veins [14]. Anatomic factors of BR PDAC include the tumor’s proximity to the superior mesenteric artery and/or celiac artery being less than 180 degrees without causing stenosis or deformity [16]. There should be no contact between the tumor and the proper hepatic artery or celiac artery, while contact with the common hepatic artery is permissible. Additionally, BR PDAC is characterized by the presence of bilateral narrowing or occlusion of the superior mesenteric vein and/or portal vein in contact with the tumor provided it does not extend beyond the inferior border of the duodenum.

Biological factors of BR PDAC include potentially resectable disease according to anatomical criteria, yet with clinical findings suggestive of distant metastases or regional lymph node involvement, and a serum carbohydrate antigen (CA) 19-9 level exceeding 500 units per milliliter. In contrast, PDAC qualifies as a LA tumor when it encases more than 50% of the arterial circumference and involves the celiac axis, superior mesenteric artery, and superior mesenteric and portal veins without distant metastasis [17,18]. From an anatomical standpoint, BR/LA PDAC tumors are relatively larger and may have invaded or become attached to the surrounding vasculature, making them challenging for surgical removal. These unique but complex phenotypical and anatomical traits of BR and LA PDAC present significant challenges in treating patients within this group.

## 2. Current Clinically Approved Therapies for Treating BR/LA PDAC

Management of BR and LA PDAC is critical. The primary treatment modalities for addressing PDAC in both BR and LA stages include chemotherapy, radiation therapy, and surgery. However, the involvement of blood vessels and the comparatively large size of tumors make these regional PDACs difficult to resect [19]. A PDAC qualifies as an LA tumor when it encompasses more than 180° of the arterial structure, enclosing over half of the total arterial circumference, whereas ≤180° arterial involvement is regarded as BR PDAC [20]. The greater the vascular involvement, the higher the risk of post-excisional complications. Other than vascular involvement, tumor size is a predictor of outcome, with larger tumors being linked to a higher risk of perineural invasion, metastasis, and a higher rate of positive lymph nodes [21,22,23]. The drugs 5-fluorouracil, leucovorin, irinotecan, and oxaliplatin (FOLFIRINOX) and gemcitabine-nab-paclitaxel (GEM-NAB) are widely used as standard chemotherapeutic agents for PDAC. Several studies have compared the efficacy, safety, and surgical optimization of FOLFIRINOX versus GEM-NAB [24,25]. Combination chemotherapy in the form of FOLFIRINOX or GEM-NAB is the primary treatment for patients with metastatic PDAC [26]. However, due to the severe toxicity associated with these treatments, they are used only in patients with good performance status [27,28]. Surgical resection is feasible, though challenging, for BR PDAC, and it increases the likelihood of early disease recurrence [29]. Approximately 73%–80% of patients with BR or LA PDAC who underwent surgical resection experienced tumor recurrence [30,31]. Chemotherapy or radiation therapy is typically used as neoadjuvant therapy prior to surgery for patients with BR or LA PDAC. However, a small percentage of patients with BR PDAC may still be good candidates for surgery without prior chemotherapy or radiation therapy [32]. Most research findings indicate improved clinical outcomes in patients who undergo surgical resection after neoadjuvant therapies compared to those who undergo surgery without neoadjuvant therapy. This includes a greater incidence of negative resection margins and higher overall survival rates. The term “margin-negative (R0) resection” is used to describe when a tumor tissue is surgically removed and no cancer cells are found on the edge of the tissue (Figure 1b) [33]. Conversely, if cancerous cells are detected at the border of the resected tissue, it is called margin-positive (R1) resection (Figure 1c), which is associated with poorer clinical outcomes [17]. Interestingly, neoadjuvant therapy with chemotherapy or radiation has often led to higher R0 resectability rates compared to adjuvant therapy in patients with resectable (R) or borderline resectable (BR) PDAC (Table 1). However, even with resection after neoadjuvant therapies, clinical outcomes remain poor, with fewer than 60% of patients achieving margin-negative (R0) resection and poor median overall survival as shown in that table. One study revealed that out of 680 patients with PDAC (267 BR and 413 LA cases) receiving FOLFIRINOX and GEM-NAB as primary chemotherapy, the overall rates of surgical resection were only 24% for BR and 9% for LA PDAC [34]. Another report highlights local tumor recurrence as the predominant post-surgical challenge for individuals with BR/LA PDAC [35]. Furthermore, large tumor size poses a threat of lymph node metastasis, vascular invasion, and elevated CA 19-9 levels, further decreasing the survival benefits of treatment [36,37,38]. TNM staging (staging based on tumor, node, and metastasis), positive lymph nodes, tumor differentiation, and CA 19-9 levels dictate the pattern and timeframe of post-operative PDAC recurrence [39]. Schorn et al. proposed that neoadjuvant therapy might reduce the likelihood of local tumor recurrence, although this efficacy is not consistently observed in managing distal tumor recurrence [40]. Moreover, altered metabolism, modification of epigenetic mechanisms, and modulation of abnormal signaling pathways were reported as the principal chemo-resistant mechanisms for PDAC [41].

The role of radiation therapy in PDAC is unclear. Historically, clinical trials have not demonstrated significant superiority of chemo-radiation therapy over chemotherapy alone in increasing median survival, with combined therapy showing more toxicity symptoms for patients [51,52]. However, advances in radiation therapy—including stereotactic body radiation therapy (SBRT), three-dimensional conformal radiation therapy, external beam radiation therapy, and proton radiation therapy—have led to better PDAC outcomes. Recent publications have demonstrated that radiation therapy alone or in combination with chemotherapy can increase overall survival and may be associated with improved outcomes [53,54]. A Phase II trial for BR PDAC showed SBRT following three cycles of GEM-NAB or FOLFIRINOX was well tolerated [55]. In this trial, out of 15 patients, 12 underwent surgery following SBRT, and 11 of those patients had R0 resection, with a median overall survival of 21 months. On the other hand, Comito et al. reported that out of 45, 32 LA PDAC patients who received neoadjuvant chemotherapy followed by stereotactic body radiotherapy had experienced a median overall survival of 19 months after diagnosis [56].

Increased tumor size in BR/LA PDAC exacerbates complications due to tumor biology [57] and is correlated with worse overall survival and disease-free survival [58]. Larger tumor size directly impacts early cancer recurrence, frequency of tumor resection, resection of other organs, and lower rates of margin-negative (R0) resection [59]. Studies strongly suggest that it is imperative to efficiently reduce the size of tumors for both BR and LA PDAC [60,61]. Reduction of tumor sizes with neoadjuvant therapy enhances the amenability for surgical resection, potentially increasing survival benefits [62]. Therefore, effective tumor size reduction through novel approaches, such as oncolytic virotherapy, could decrease the risk of metastasis, enhance the predictability of tumor staging and resectability, minimize vascular involvement, and reduce positive lymph nodes.

## 3. Oncolytic Virotherapy—An Emerging Cancer Immunotherapy for BR/LA PDAC

Oncolytic virotherapy has garnered interest as a potential category of immunotherapy. Oncolytic viruses (OVs) have the natural or induced ability to selectively infect tumor cells, replicate inside them, boost the antitumor immune response, and elicit apoptosis to kill cancer cells, thereby downsizing tumors [63]. Because most cancer cells have defective interferon or protein kinase R signaling (PKR), OVs can bypass antiviral mechanisms to infect and replicate within these cells [64]. Tumor cells are notorious for low expression of apoptotic enzymes, but OVs typically induce apoptosis in those cells by manipulating signaling pathways, including, but not limited to, PI3K/Akt, Daxx, PKR, Wnt, Ras, or Fas [65,66]. In addition to oncolysis, OVs also stimulate innate and adaptive antitumor immune responses by releasing damage-associated molecular patterns, such as heat shock proteins, high-mobility group box 1b proteins, viral proteins, and tumor-specific antigens [66]. Consequently, natural killer (NK) cells, macrophages, dendritic cells, and cytotoxic T cells are mobilized to the infection site, aiming to eradicate the tumor cells, leading to a cross-reacting immune response against the tumor [66,67]. A study indicates that immune response against tumor antigens was potentiated after patients with multiple myeloma were treated with oncolytic measles virus [68].

OVs utilize specific cell surface receptors or protein molecules to attach and infect host cells. For example, herpes simplex virus 1 binds to the herpesvirus entry mediator, measles virus binds to CD46, echovirus binds to integrin α2β1, vesicular stomatitis virus binds to the LDL receptor, and coxsackievirus binds to intercellular adhesion molecule 1 [69]. Each OV leverages molecular and cellular differences between healthy and cancer cells. TVEC-1, the first FDA-approved herpes simplex virus 1-based OV, has deletions in the viral infected cell protein ICP34.5 neurovirulence gene and ICP47, increasing tumor selectivity and reducing pathogenicity [70,71]. Other Ovs, such as parvovirus, adenovirus, and reovirus, show specificity to cancer cells with mutated or non-functional p53 [69]. Vaccinia and reovirus are specific to cancer cells with hyperactive Ras signaling [69]. Vesicular stomatitis virus is favorable in treating cancer cells that lack functional antiviral interferon signaling [72].

The success of oncolytic virotherapy, like any cancer immunotherapy, relies on the ability of virus particles to reach cancer cells, kill tumor cells, and recruit antitumor immune molecules within the tumor region. The TME, with its cellular and acellular components, creates an immunosuppressive dense stroma that acts as a physical barrier to OV and immune cell infiltration [73]. Although some studies suggested that depletion of stromal αSMA (+) myofibroblasts—a cellular component of PDAC TME—is associated with disease progression [74] and the invasive nature of PDAC [75], preclinical and clinical data indicate that accumulation of hyaluronic acid and collagen—significant acellular components of the PDAC TME—is associated with aggressive metastatic disease, drug resistance, and poor prognosis [12]. Therefore, it is necessary to cautiously but effectively downregulate or deplete PDAC stromal components to improve the intratumoral bioavailability of OVs as well as cytotoxic T cell infiltration into the TME and induce a response in tumors.

## 4. Intricate Interplay of Acellular Components of the TME in PDAC Development and Treatment Resistance

The extracellular matrix of the stroma plays a pivotal role in BR/LA PDAC progression and creates a barrier against anticancer drug delivery. Carbohydrates, including polysaccharides and glycoproteins, along with proteins such as proteoglycans and others, collectively form an extracellular matrix which is characterized by the presence of both interstitial matrix and basement membrane [76]. While the interstitial matrix accommodates collagens type I, II, III, V, XI, XXIV, and XXVII, the basement membrane contains type IV and type VIII collagen. Moreover, hyaluronic acid, integrin, fibronectin, vitronectin, laminin, and actin are some of the important components of the PDAC extracellular matrix that impart stiffness to the TME and assist in PDAC propagation [77,78]. These acellular components of the TME build up stromal resistance that impede drug and antitumor immune cell penetration to the tumor site (Figure 2).

Hyaluronic acid, a glycosaminoglycan, is one of the most prevalent substances in the PDAC extracellular matrix. Hyaluronic acid synthases play a primary role in synthesizing hyaluronic acid [79]. High levels of hyaluronic acid are also indicative of a poor prognosis of PDAC [80]. Surprisingly, metastasis might be associated with higher blood hyaluronic acid concentrations [80]. Healthy individuals and patients with benign conditions have lower serum hyaluronic acid levels than patients with PDAC [81]. In PDAC, several components of the extracellular matrix, such as laminin, collagens, and osteopontin, are overexpressed and interact with CD44, the principal cell surface receptor for hyaluronic acid [82,83]. Through this receptor, hyaluronic acid actively contributes to cancer progression and invasion [79]. High levels of hyaluronic acid are linked to an increase in tumor interstitial pressure by capturing water, establishing a physical barrier that severely limits the penetration and delivery of drugs into tumor tissues [84]. Interestingly, post-surgical survival was observed to be inversely proportional to the hyaluronic acid level in PDAC stroma [85].

Collagen, one of the major acellular components of PDAC stroma, impacts PDAC progression, metastasis, plasticity, and chemo-resistance [86,87,88,89]. During the transformation of normal pancreatic tissue to PDAC, type I and type III collagen masses increase almost 2.6-fold [90]. Additionally, elevated expression of type IV collagen is observed in pancreatic cancer tissue in the immediate vicinity of cancer cells within the tumor stroma [91]. Collagen type VI protein is overexpressed in the highly metastatic subpopulation of BxPC-3, which is a human pancreatic cancer cell line [92].

Laminin is crucial in preserving the activity and functionality of the basement membrane, standing out as one of the most enriched non-collagenous heterotrimer proteins within this membrane [79]. Laminin-332 serves as a mediator in PDAC, influencing key processes such as proliferation, apoptosis, invasion, migration, and epithelial-to-mesenchymal transition [93]. Laminin is encoded by the laminin subunit alpha 3 (LAMA3), laminin subunit beta 3 (LAMB3), and laminin subunit gamma 2 (LAMC2) genes and functions in the epithelial-to-mesenchymal transition, invasion, and proliferation of PDAC [93]. Laminin is associated with activating the PI3K/Akt signaling pathway to facilitate PDAC progression [94]. Furthermore, LAMA3 and LAMC2 show promise as potential therapeutic targets and can also be repurposed as markers for PDAC prognosis [95].

Integrins act as cell surface receptors for laminin, collagen, fibronectin, and other glycoproteins, and their activation recruits the adhesome network [78,96]. A bidirectional collaboration between cancer cells and the stroma is established by integrin signaling, which usually stimulates the actin cytoskeleton to interact with the extracellular matrix [97]. Moreover, PDAC progression and proliferation have been linked to the action of integrin α5β6 [98]. One interesting study showed that cancer cell migration occurs along fibronectin fibers with the help of integrin α5β1 [99]. Additionally, the release of cancer cells from cell bundles can be facilitated by the proteolytic action within the extracellular matrix driven by integrin via expressing matrix metalloproteinases [100].

In normal tissue, fibronectin in the extracellular matrix is necessary for tissue regeneration and wound healing [101]. Fibronectin is also active in tumors, promoting PDAC cell invasion, proliferation, and metastasis [102,103]. These roles of fibronectin, along with laminin and collagen, are linked to the production of reactive oxygen species. Overproduction of reactive oxygen species mediates oxidation of membrane phospholipids and can lead to cell damage and tissue necrosis [104]. Elevated reactive oxygen species level is implicated in the initial stages of cancer initiation and progression [105]. Fibronectin-stimulated production of reactive oxygen species was shown to confer higher cancer cell survival in cocultures of PDAC cell lines such as MIA PaCa-2, PANC-1, and Capan-1 [106]. Therefore, efforts directed at addressing the acellular components within the stroma of the TME in PDAC represent a promising avenue for future therapeutic advancements.

## 5. Targeting Stromal Acellular Components in PDAC by Stroma-Depleting Enzymes

Several naturally occurring and synthetic enzymes have the potential to deplete components of the PDAC stroma to improve the penetration of anticancer therapies. Some of these enzymes have been the subject of clinical studies, while others remain to be tested. Numerous scientific studies have documented the targeting of acellular components within the TME as a strategy to enhance the delivery of anticancer drugs. The most focused and clinically advanced approach targets hyaluronic acid with hyaluronidase enzyme (PH20). In vivo data suggested that oncolytic vaccinia encoding a soluble version of hyaluronidase Hyal1 demonstrated beneficial therapeutic outcomes when combined with chemotherapy, anti-CD47 therapy, or anti-PD-1 therapy [107]. A Phase I clinical trial was conducted involving an OV coding for PH20 alone (Part I), concurrent with chemotherapy (Part II), or as neoadjuvant therapy followed by nab-paclitaxel plus gemcitabine (Part III) in patients with advanced PDAC [108]. Results of the trial demonstrated increased levels of interferon-γ, soluble lymphocyte activation gene-3 expression, interleukin (IL)-6, and IL-10, but some patients also experienced dose-limiting toxicities, such as increased grade 4 aspartate aminotransferase level, grade 4 febrile neutropenia, and grade 5 thrombocytopenia plus enterocolitis.

Another Phase I clinical trial illustrated that polyethylene glycolated recombinant human hyaluronidase alfa (PEGPH20) showed compelling antitumor efficacy and was tolerable up to a dose of 3 μg kg^−1^ administered twice per week, although 29% of patients experienced thromboembolic effects [109]. A Phase Ib trial demonstrated that patients with higher hyaluronic acid content in their tumor stroma responded well to combination therapy of PEGPH20 and gemcitabine, with improved progression-free survival and overall survival [110]. Subsequently, a Phase II trial demonstrated that PEGPH20 increased progression-free survival and overall survival in patients with high hyaluronic acid levels [111]. These promising findings paved the way for a larger randomized Phase III study. However, the results of this trial were disappointing, as they showed no significant difference in progression-free survival and overall survival between patients who received PEGPH20 in combination with GEM-NAB and those who received placebo plus GEM-NAB [112].

Achieving consistent distribution of proteolytic enzymes through intratumoral administration is a challenging task, increasing the risk of off-target delivery [113]. Proper drug delivery becomes even more complicated in solid tumors containing dense stroma [114]. On the other hand, systemic administration of such drugs poses the risk of adverse effects. In most clinical studies, the adverse events associated with intravenous administration of PEGPH20 included muscle spasms, hyponatremia, and thromboembolism [112,115]. One study reported that PEGPH20 could potentially enhance the sensitivity of pancreatic cancer to radiation, but its effectiveness may be limited to tumors exhibiting significant accumulation of hyaluronic acid [116]. Unfortunately, concurrent administration of PEGPH20 with FOLFIRINOX in a non-specific patient cohort led to unfavorable overall survival outcomes, with increased rates of grade ≥ 3 toxicity [115]. To address concerns regarding the potential for metastasis after PH20 treatment, a study demonstrated that degrading extracellular matrix components of tumors by local PH20 expression neither contributed to malignant cell invasion in vitro nor did it facilitate metastasis in vivo [107].

In an attempt to overcome the challenge of the impediment of drug penetration associated with collagen fibers, a study was carried out using liposomes that encapsulate collagenase, known as collagozome [117]. The intravenous injection of collagozomes required approximately 8 h to achieve around 1% of the total dose within the pancreas, leading to an enhancement in drug penetration. The study concluded that pretreatment with collagozomes followed by paclitaxel micelles caused an 87% greater reduction in tumor size compared to tumors pretreated with empty liposomes followed by paclitaxel treatment. A study by Ebelt et al. suggested that *Salmonella typhimurium* expressing bacterial collagenase improved treatment outcomes by decreasing immunosuppressive subsets and causing tumor regression [118]. However, the study lacks comprehensive toxicological data. In addition, sonodynamic therapy is another promising therapeutic approach for the treatment of pancreatic cancer. It uses ultrasound to trigger the production of reactive oxygen species and elicit tumor cell hypoxia and apoptosis. Nanoparticles containing hollow titanium dioxide conjugated with collagenase improved visualization for sonodynamic therapy ultrasound signal and increased generation of reactive oxygen species needed for tumor hypoxia [119]. However, a study concluded that type IV collagenase with an isoform of 92 kDa influences cancer invasion [120].

Bromelain, a proteolytic enzyme extracted from pineapples, has shown efficacy in digesting tumor stroma. Several studies have demonstrated the effect of bromelain against different cancer types ranging from breast cancer to colon cancer [121,122]. A nanoparticle made of bromelain-immobilized and lactobionic-acid-modified chitosan demonstrated higher chemotherapeutic drug biodistribution and antitumor efficacy in monolayer cells and three-dimensional cell spheroids [123]. Another study from the same group confirmed that a pH-dependent bromelain nanoparticle could enhance the infiltration of chemotherapeutic agents within tumor [124]. Parodi et al. reported that mesoporous silica nanoparticles conjugated with bromelain increased the penetration of the nanoparticle conjugate within cancer parenchyma [125]. The efficacy of bromelain in effectively digesting stromal components and increasing heat production from photothermal therapy has also been studied [126]. The findings of this study indicate that semiconducting polymer nanoenzymes containing bromelain digested collagen efficiently and increased the activity of photothermal therapy.

Research has revealed that relaxin plays a role in the digestion of various stromal components, including collagen [127]. Previously, relaxin was known to exert an anti-fibrotic effect, inhibiting the generation and secretion of collagen [128,129]. One interesting study showed that an oncolytic adenovirus expressing relaxin potentiated the cytotoxic effect of GEM even with its subtherapeutic dose [130]. The combination of these two agents decreased the expression levels of collagen, fibronectin, and elastin in both tumor spheroid and xenograft tumor models. In another study, serelaxin, the recombinant form of human relaxin-2, demonstrates both improved anti-fibrotic capability by inhibiting collagen I and III and endothelial to mesenchymal transition [127,131].

Cysteine protease enzymes such as papain and papain-like cathepsin-K and cathepsin-B possess the potential to enhance the degradation of collagen [132]. It showed that papain specifically targets the Glutamine-Glutamine-Aspartic acid motif for degrading its substrate, whereas collagen contains only the Glutamine-Glutamine motif. Cathepsin-L, a closely related homolog of cathepsin-K, has demonstrated efficient cleavage of type I collagen within the triple helix region despite the absence of proline specificity [133]. While releasing fragments from reconstituted fibrils of FITC-labeled collagen, cathepsin-L demonstrated four times less potency than cathepsin-K. Atomic force microscopy evidence has shown that papain gel partially degrades intact, non-mineralized type I collagen fibrils [134].

Several investigations have concentrated on the capability of purified human neutrophil elastase to cleave type I collagen. Neutrophil elastase has collagenolytic activity against soluble human, bovine, and rat type I collagen while cleaving reconstituted and radio-labeled type I collagen fibrils [135]. Mainardi et al. demonstrated polymorphonuclear leukocyte elastase could cleave native human liver type III collagen by cleavage within triple helix regions, although it could not cleave human type I collagen [136]. This result is supported by another study, which illustrated that triple helical regions of type I collagen are kept unharmed by neutrophil elastase, but the enzyme can produce collagen fragments by cleaving triple helical human type III collagen [137]. Furthermore, one interesting study showed that neutrophil elastase could break apart chick-derived native cartilage collagen type X with three cleavages [138].

Several collagenolytic serine proteases from the subtilisin family, sourced from environmental or pathogenic microorganisms, have been reported. S8 collagenolytic protease, sourced from *Geobacillus collagenovorans* MO-1, stands out for its thermostability [139]. Interestingly, this enzyme, characterized by a collagen-binding domain located at its C-terminal, exhibits specificity in targeting various cleavage sites on collagen [140]. Collagenolytic protease MCP-01 has demonstrated a gradual release of single fibrils from collagen fibers through the hydrolysis of proteoglycans and telopeptides within the collagen fibers, liberating collagen monomers from these fibrils by hydrolyzing proteoglycans and telopeptides within the collagen fibers [141]. Zhang et al. revealed that purified matured collagenolytic protease from thermophilic *Brevibacillus* sp. WF146 contains a subtilisin-like catalytic domain and showcased notable activity against insoluble type I collagen and azocoll, accompanied by limited gelatinolytic activity. The protease demonstrated a distinct preference for arginine and glycine [142].

It is important to note that numerous natural or synthetic enzymes capable of degrading various PDAC stromal components exist. These enzymes present a potential to make solid tumors more permeable to anticancer therapies. However, comprehensive investigations are necessary to thoroughly examine the effects, effectiveness, feasibility, and safety of proteolytic enzymes in degrading stromal components of PDAC to enhance the benefit of anticancer therapy.

## 6. Are Oncolytic Viruses Expressing Stroma-Degrading Proteolytic Enzymes the Next Game-Changer?

A major issue in BR/LA PDAC resection is the likelihood of encountering positive margins, which can lead to tumor recurrence [35]. While many clinical studies have shown encouraging results of neoadjuvant or adjuvant chemotherapy or radiation therapy on the resection of BR/LA PDAC, surgeons and oncologists face several obstacles. These include limited therapeutic success, loss of curative resection timelines, ambiguity of anatomical tumor staging, presence of positive margins in resected tissue, risk of metastasis, cancer recurrence, and the association of vasculature with PDAC tumors [32,143,144,145]. As a result, there is a pressing need for novel therapies alone or in conjunction with chemo-radiation therapy to downstage BR/LA PDAC. As previously mentioned, Phase I and Phase II clinical trials utilizing PH20 demonstrated encouraging results, although these enzymes failed to show an overall efficacy in subsequent Phase III clinical trials [108,112,115]. It is hypothesized that utilizing optimized stroma-degrading proteolytic enzymes incorporated into and expressed via OVs could decrease tumor size through accelerating direct oncolysis of cancer cells and reduction of stromal components. A preclinical study with an oncolytic adenovirus expressing PH20 showed better results in decreasing tumor volume in a melanoma model compared to the oncolytic adenovirus alone [146]. Another study by Bazan-Peregrino et al. suggested that the infiltration of anti-PD-L1 antibody was significantly enhanced within the tumor after treatment with PH20-expressing OVs compared to OVs lacking PH20 [147]. They further demonstrated that OVs expressing PH20 exhibited superior performance in controlling tumor size and delaying tumor progression compared to chemotherapy or OVs lacking PH20.

The process of genetic engineering of oncolytic virus is scientifically validated [148]. It involves two main steps: cloning and viral rescue (Figure 3). This figure schematically depicts the molecular engineering and rescue of oncolytic vesicular stomatitis virus (VSV) expressing proteolytic enzymes. Such OVs are expected to increase the possibility of margin-negative (R0) resection for both BR and LA PDAC tumors. The aim of initiating studies with such OVs to target the PDAC stroma is to reduce desmoplasia, thereby increasing the likelihood of R0 resection of BR/LA PDAC (Figure 4). An intratumoral injection of OVs expressing proteolytic enzymes will provide local efficacy within the tumor microenvironment and should bypass systemic toxicity. This approach is expected to lead to higher tumor cell death and tumor shrinkage via (1) direct virus-induced destruction of cancer cells (oncolysis) and (2) improved infiltration of immune cells into the tumor. The anticipated outcomes are reduced tumor burden, delayed tumor growth, and the potential extension of survival in patients with BR/LA PDAC, marking a significant step forward in treating this challenging cancer subset.

Preoperative tumor downstaging therapy for patients with BR/LA PDAC has increased the survival rate of these patients to a level closer to that of patients with early-stage resectable tumors [149]. Patients with tumor size greater than >2 cm often have lymph node metastasis with a combination of perineural, vascular, and lymph vessel invasion [57]. As a result, downsizing tumor size with OVs-mediated expression of proteolytic enzymes for stroma degradation may ensure the reduction or absence of vascular involvement within the tumor and surrounding tumor framework, resulting in a reduced risk of metastasis. Furthermore, OV-mediated tumor cell death can prime and activate tumor antigen-specific T cell responses, which can target not only the primary tumor but also pockets of minimal residual disease that go undetected by imaging assessment or surgery. OV-based therapeutic approaches may also be combined with other therapeutic modalities such as chemo-radiation therapy or immunotherapy, synergizing therapeutic outcomes.

About one-fourth of patients initially present with BR and/or LA tumors that cannot be surgically removed. Despite lacking metastatic disease, converting these tumors to a resectable state through treatment could potentially improve clinical outcomes significantly. Therefore, focusing on this patient group, which represents a substantial portion, offers a promising opportunity for impactful clinical advancements. This approach does not disregard its relevance in other clinical scenarios but underscores an area where clinical needs remain unmet. The primary goal remains shrinking tumors to facilitate other treatment modalities. While addressing unresectable tumors is crucial, any situation where a positive response translates into meaningful clinical benefits warrants exploration. However, before incorporating proteolytic entities into OVs to combat PDAC, there is a need for intense scientific efforts to identify and characterize novel candidate enzymes with proteolytic activity along with evaluating their efficacy and safety.

## 7. Conclusions

BR and LA PDAC pose formidable challenges in clinical management. Complications such as vessel involvement, a high ratio of positive lymph nodes, large tumor size, and unpredictable resection margins contribute to the recurrence or metastasis of BR/LA PDAC after surgical resection. Oncolytic virotherapy has shown promise in enhancing cancer cell destruction, releasing neoantigens, and recruiting antitumor immune cells, thereby transforming a pro-tumorigenic environment into an antitumorigenic one. The dense desmoplastic stroma in PDAC exacerbates resistance which limits the penetration of therapeutic agents and immune cells, thus reducing the efficacy of anticancer treatments. Stroma-degrading proteolytic enzymes can potentially soften this rigid stromal structure, thereby enhancing the effectiveness of other therapies. Incorporating stroma-depleting enzymes into OVs, either alone or in combination with standard therapies like chemotherapy or radiation therapy, holds promise for future PDAC treatment strategies. However, the potential survival benefits, toxicological profiles, and risks of metastasis associated with this approach must be thoroughly investigated to ascertain the clinical efficacy of OVs expressing proteolytic enzymes in treating BR/LA PDAC.

## Figures and Tables

**Figure 1 biomedicines-12-01596-f001:**
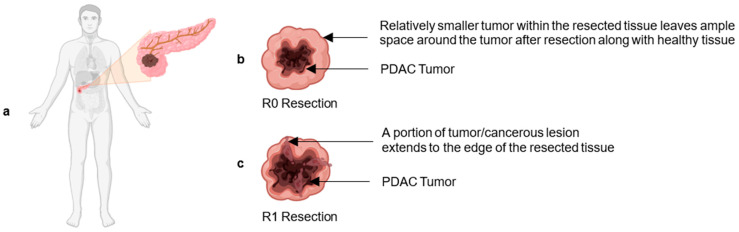
(**a**) A male patient with PDAC. (**b**) The edge of the surgically removed tissue is free of cancer lesions, indicating R0 resection. (**c**) The edge of the surgically removed tissue contains cancerous lesions, resulting in R1 resection.

**Figure 2 biomedicines-12-01596-f002:**
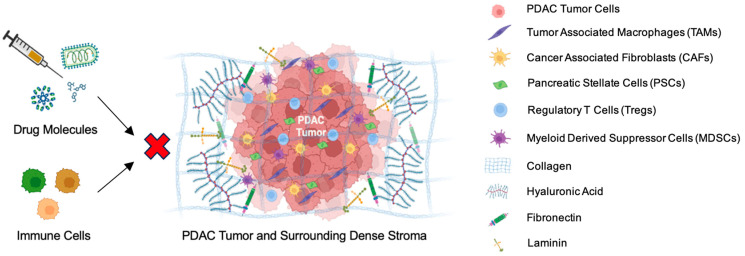
PDAC tumor microenvironment is composed of immunosuppressive intricate structures of cellular and acellular components like TAMs, CAFs, PSCs, Tregs, MDSCs, collagen, hyaluronic acid, fibronectin, and laminin. This complex structure establishes a dense stromal barrier, obstructing the entry of drug molecules like chemotherapy or immunotherapies, along with anti-tumor immune cells.

**Figure 3 biomedicines-12-01596-f003:**
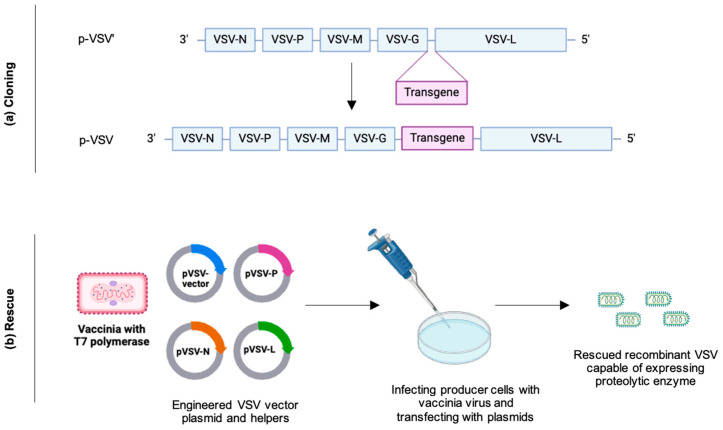
Schematic of molecular engineering of oncolytic VSV virus expressing proteolytic enzymes. (**a**) Cloning of VSV plasmid (p-VSV′) is done by inserting transgene (encoding an enzyme) in between VSV G and L genes, giving rise to a plasmid of engineered VSV variant (p-VSV) containing a gene that expresses proteolytic enzyme. (**b**) Producer cells (example: BHK21, Vero) are infected with vaccinia virus with T7 polymerase along with transfection with p-VSV vector and helper plasmids (pVSV-P, pVSV-N, pVSV-L). This step is called viral rescue. The supernatant is filtered and is used to infect a new batch of cells that ultimately yields VSV particles capable of expressing stroma-depleting proteolytic enzyme.

**Figure 4 biomedicines-12-01596-f004:**
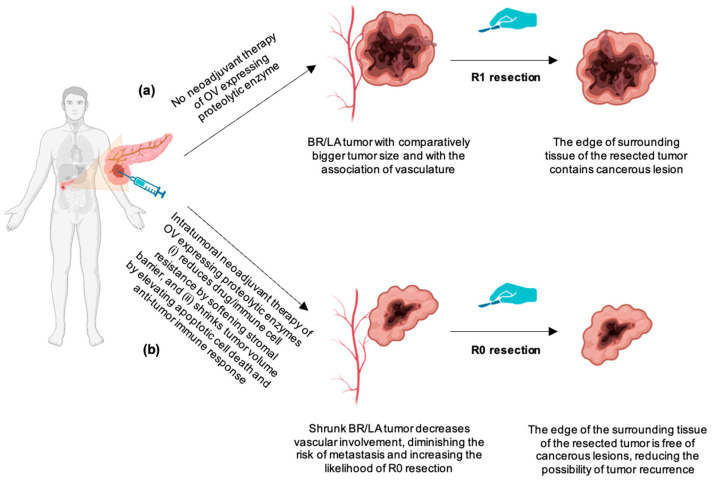
(**a**) The increased size of a BR or LA PDAC tumor raises the probability of a cancerous lesion persisting even post-surgery, a condition referred to as R1 resection. Further complicating the situation, the vascular involvement also increases the risk of metastasis. (**b**) Intratumoral injection of OV expressing proteolytic enzyme in BR/LA PDAC tumors decreases the stromal rigidity and increases drug/immune cell infiltration, thereby increasing tumor cell death and boosting the immune response against the tumor. Consequently, these events may lead to the likelihood of R0 resection with no vascular involvement, metastasis, or tumor recurrence.

**Table 1 biomedicines-12-01596-t001:** Summaries of studies on the effect of neoadjuvant therapy, adjuvant therapy or upfront surgery in clinical outcomes for patients with PDAC.

PDAC Type	Cohort	Therapy	R0 Resectability (%)	MS/OS (Months)	Source
BR	134	FOLFIRINOX or modified FOLFIRINOX followed by SBRT, then surgery	64%	22	[42]
BR	27	Gemcitabine and external beam radiation followed by surgery	51.8%	21	[43]
	23	Upfront surgery followed by chemoradiation	26.1%	12	
R and BR	119	Gemcitabine followed by radiotherapy, then surgery	72%	16.0	[44]
	127	Immediate surgery followed by adjuvant gemcitabine	40%	14.3	
BR and LA	135	Gemcitabine monotherapy, gemcitabine-capecitabine, and gemcitabine-erlotinib or FOLFIRINOX followed by surgery	NR	17.1	[45]
	359	Upfront surgery	NR	7.1	
BR and LA	34	Low-dose gemcitabine and wide irradiation area	47.1%	39.0	[46]
BR and LA	30	Dose escalated radiotherapy (helical tomotherapy) with concurrent chemotherapy (gemcitabine)	39.0% (BR)	11.8 (LA), 17.3 (BR)	[47]
BR and LA	253	FOLFIRINOX with or without radiotherapy	39.4%	NR	[48]
BR and LA	20	Gemcitabine and nab-paclitaxel followed by RT, then surgery	20%	NR	[49]
LA	12	chemoradiotherapy using gemcitabine plus nab-paclitaxel	50%	NR	[50]

R = resectable, MS = median survival, OS = overall survival, SBRT = stereotactic body radiation therapy, nab = nanoparticle-albumin-bound, NR = not reported.

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
