# Peer review of "Enhancing Neoadjuvant Virotherapy’s Effectiveness by Targeting Stroma to Improve Resectability in Pancreatic Cancer"

_biomedicines, 2024, doi:10.3390/biomedicines12071596_

Round 1

Reviewer 1 Report

Comments and Suggestions for Authors

The present review provides a comprehensive analysis of the therapeutic advantages offered by oncolytic viruses in enhancing pancreatic cancer treatment. Overall, this well-crafted article aligns seamlessly with the journal's thematic focus. I am pleased to recommend that the manuscript be accepted pending minor revisions.

1. This review emphasizes the significance of the tumor microenvironment (TME) in the progression of PDAC, particularly highlighting the immunosuppressive characteristics of TME. This includes examining various cell types such as pancreatic stellate cells, tumor-associated macrophages, cancer-associated fibroblasts, regulatory T cells, myeloid suppressor cells, and extracellular matrix proteins like hyaluronic acid and collagen. Additionally, it is suggested that the author can incorporate TEM-related introduction within the background section. The following literature can be cited to clarify the role of TEM:

[1] Hartupee, C.; Nagalo, B. M.; Chabu, C. Y.; Tesfay, M. Z.; Coleman-Barnett, J.; West, J. T.; Moaven, O., Pancreatic cancer tumor microenvironment is a major therapeutic barrier and target. Frontiers in Immunology 2024, 15, 1287459. DOI: 10.3389/fimmu.2024.1287459

[2] Gu, A.; Li, J.; Qiu, S.; Hao, S.; Yue, Z.-Y.; Zhai, S.; Li, M.-Y.; Liu, Y., Pancreatic cancer environment: from patient-derived models to single-cell omics. Molecular Omics 2024, 20 (4), 220–233. DOI: 10.1039/D3MO00250K

[3] Sherman, M. H.; Beatty, G. L., Tumor Microenvironment in Pancreatic Cancer Pathogenesis and Therapeutic Resistance. Annual Review of Pathology-Mechanisms of Disease 2023, 18, 123-148. DOI: 10.1146/annurev-pathmechdis-031621-024600

[4] Palma, A. M.;  Bushnell, G. G.;  Wicha, M. S.; Gogna, R., Tumor microenvironment interactions with cancer stem cells in pancreatic ductal adenocarcinoma. Advances in cancer research 2023, 159, 343-372. DOI: 10.1016/bs.acr.2023.02.007

2. The end of the introduction should provide readers with an overview of the review's content and the author's purpose for writing, enhancing reader comprehension for subsequent sections.

3. The author should add the conclusion part to summarize elucidating the benefits of employing oncolytic viruses in the treatment of pancreatic cancer and outlining the potential for future advancements.

4. The author should meticulously review the manuscript for grammatical and other issues to ensure the review's quality. All words requiring italics must be correctly formatted. Additionally, all abbreviations should be defined upon first usage and consistently employed throughout subsequent texts.

5. The references should be meticulously reviewed to ensure adherence to the journal's requirements. It is imperative that all journal names are accurately abbreviated.

Author Response

Dear Reviewer 1, please see the attachment.

Reviewer 2 Report

Comments and Suggestions for Authors

This is nicely written review of the potential use of oncolytic viruses to degrade stroma of pancreatic tumors to improve surgery outcomes.

The paper deals in detail with the different components of the stroma and strategies used to degrade them.

Only a few minor comment to improve the paper:

The paper describes many options to degrade stroma , being Oncolytic viruses one of those. The use of OVs seems speculative based on theoretical advantages.  Is there any evidence that OVs are better than soluble enzymes to degrade stroma?

The emphasis of the paper is preoperative (neoadjuvant) downstaging to improve surgical resection of PDAC. It is not clear why authors consider this more relevant than adjuvant therapy. Would they favor neoadjuvant trials instead of trials in non-resectable cases? If an OV fails to show benefit in patients with non-resectable PDAC should still be tried in the neoadjuvant setting?

Page 5, Lane 189 “As a result, a cross-reacting immune response is mounted against the tumor 64,65” .  To gauge the real potential of virotherapy, it is crucial to understand the evidence of such immune responses against tumor antigens elicited by viruses, which otherwise clearly elicit anti-viral responses. A deeper immunocentric analysis could be provided. Has this antitumor immune responses been described ever in PDAC patients? In other patients? Or only has been observed in mouse models with strong tumor neoepitopes, where tumors are engraphted and therefore a lower immune tolerance to tumor neoepitopes is present?

Page 6 lane 237: “Surprisingly, metastasis might be associated with higher blood hyaluronic acid concentration 78 , and healthy pancreases have been observed to contain lower serum hyaluronic acid levels than those with PDAC 79”  This sentence seems to refer to healthy persons or patients, but as currently written makes no sense.

Page 6 “With this receptor, hyaluronic acid actively participates in cancer progression and invasion 77 .”  Please check correct reference numbers. This sentence goes with reference 76 not 77…and so on (the following 82 should be 77…revise carefully all the paper).

Author Response

Dear Reviewer 2, please see the attachment.

Round 2

Reviewer 1 Report

Comments and Suggestions for Authors

Accept in present form

Reviewer 2 Report

Comments and Suggestions for Authors

None